# Evaluation of Formalin-Inactivated Vaccine Efficacy against Red Seabream Iridovirus (RSIV) in Laboratory and Field Conditions

**DOI:** 10.3390/vaccines12060680

**Published:** 2024-06-19

**Authors:** Joon-Gyu Min, Guk-Hyun Kim, Chong-Han Kim, Woo-Ju Kwon, Hyun-Do Jeong, Kwang-Il Kim

**Affiliations:** 1Department of Aquatic Life Medicine, Pukyong National University, Busan 48513, Republic of Korea; cdmin0621@gmail.com (J.-G.M.); rnrgus8055@gmail.com (G.-H.K.); jeonghd@pknu.ac.kr (H.-D.J.); 2Vaccine Research Institute, Woogene B&G Co., Ltd., Seoul 07299, Republic of Korea; chonghan2004@woogenebng.com (C.-H.K.); cosmos0021@woogenebng.com (W.-J.K.)

**Keywords:** RSIV, vaccine, field trial, formalin-inactivated vaccine, rock bream

## Abstract

Red seabream iridovirus (RSIV) is a major cause of marine fish mortality in Korea, with no effective vaccine available since its first occurrence in the 1990s. This study evaluated the efficacy of a formalin-killed vaccine against RSIV in rock bream under laboratory and field conditions. For the field trial, a total of 103,200 rock bream from two commercial marine cage-cultured farms in Southern Korea were vaccinated. Farm A vaccinated 31,100 fish in July 2020 and monitored them for 18 weeks, while farm B vaccinated 30,700 fish in August 2020 and monitored them for 12 weeks. At farm A, where there was no RSIV infection, the vaccine efficacy was assessed in the lab, showing a relative percentage of survival (RPS) ranging from 40% to 80%. At farm B, where natural RSIV infections occurred, cumulative mortality rates were 36.43% in the vaccinated group and 80.32% in the control group, resulting in an RPS of 54.67%. The RSIV-infectious status and neutralizing antibody titers in serum mirrored the cumulative mortality results. This study demonstrates that the formalin-killed vaccine effectively prevents RSIV in cage-cultured rock bream under both laboratory and field conditions.

## 1. Introduction

*Megalocytiviruses*, a genus in the family Iridoviridae, comprise a group of double-stranded DNA viruses encapsulated within an icosahedral capsid. They cause significant diseases in teleost fish, particularly red seabream iridoviral disease (RSIVD) and scale-drop disease [1,2]. Among the three genotypes (red seabream iridovirus, infectious spleen and kidney virus, and turbot reddish body iridovirus) based on phylogenetic analysis of major capsid protein (MCP) and adenosine triphosphatase genes [1], RSIV has been a predominant pathogen in marine fish in Korea since the 1990s, causing significant mortality in species such as rock bream (*Oplegnathus fasciatus*) [3,4,5], one of the valuable marine fish species in Korea [6]. RSIV frequently occurs when water temperatures range from 23 °C to 27 °C, which are typically observed in South Korea between August and October. This period enhances the proliferation and transmission of the RSIV pathogen, leading to mass mortality in aquaculture farms [7].

Vaccination is a key strategy for preventing diseases in fish by enhancing the immune response. Vaccines provide long-term protection against specific viral infections, thereby decreasing mortality rates and the associated economic losses for fish farms. They also reduce the risk of severe outbreaks, contributing to more sustainable and efficient aquaculture practices. In particular, inactivated vaccines are considered highly efficient, and their ability to be produced at a low cost makes them ideal for use in the aquaculture industry [8]. Despite various measures, including the use of DNA vaccines and immunostimulants, achieving effective control of RBIV remains challenging [9]. Although two vaccines are commercially available to prevent *megalocytivirus* infection [10], they do not include rock bream as a target species for vaccination. The Biken vaccine, which was produced based on the RSIV subtype-I strain, was introduced and used in rock bream in Korea; however, it did not show sufficient efficacy. In addition, the protection of fish species belonging to the genus *Oplegnathus* by vaccination is difficult because of the high susceptibility of these species to RSIV infection [1]. Thus, in Korea, where RSIV-subtype 2 is predominantly distributed [5], such genotype-based vaccines should be developed and commercialized.

In our previous study [11], RSIV vaccine efficacy against rock bream, the species most susceptible to RSIV infection, was evaluated under laboratory conditions based on factors including antigen dose, inactivation methods, adjuvant types, and water temperature. Among the factors tested, rock bream vaccinated with a high-dose, ultracentrifuged *megalocytivirus* vaccine (Ultra HSCMV, 7.0 × 10^10^ copies/mL) showed significantly higher survival, with only 10% cumulative mortality, compared to those given relatively low-dose vaccines (1.0 × 10^10^ copies/mL, 1.0 × 10^9^ copies/mL, and 1.0 × 10^8^ copies/mL), which had cumulative mortality rates of 48.3 ± 7.6%, 75.0 ± 5.0%, and 100.0 ± 0.0%, respectively, indicating an antigen dose-dependent vaccine efficacy. However, even if a vaccine is effective under laboratory conditions, demonstrating its efficacy in the field is challenging due to numerous variables. To address this, our study aimed to evaluate the practical applicability of the developed formalin-inactivated RSIV vaccine in rock breams under field conditions. By conducting field trials in different commercial fish farms and assessing various parameters, such as growth performance, spleen index, and neutralizing antibody levels, this study aimed to identify the most effective strategies for protecting rock bream against RSIV infection. These findings will contribute to improving disease management strategies in aquaculture, ultimately enhancing the sustainability and productivity of the aquaculture industry.

## 2. Materials and Methods

### 2.1. Cell Culture

The virus was cultured in the *Pagrus major* fin (PMF) cell line as reported by [12] to yield high-titer cultures. PMF cell lines were seeded into each 850 cm^2^ roller bottle along with 180 mL of Leibowitz’s (L-15) media (Sigma Aldrich, Burlington, MA, USA) containing 10% fetal bovine serum (Gibco, New York, NY, USA), 1% antibiotic-antimycotic solution (Gibco, New York, NY, USA), and 0.5% black seabream (*Acanthopagrus schlegelii*) blood serum. The bottles were then placed in a roller apparatus at 25 °C with a speed maintained at 1 rpm during the incubation.

### 2.2. Virus Propagation and Vaccine Preparation

The iridovirus sachun-1 (IVS-1) strain, belonging to RSIV subtype 2 and isolated from rock bream with RSIVD, was used as an antigen. Once a 100% monolayer of PMF cells was confirmed, the viral stock was inoculated at a multiplicity of infection (MOI) of 100. Seven days post-inoculation, 80% of the medium was replaced and cytopathic effects (CPE) were monitored. At 12–15 days post-inoculation, when the virus copies peaked, the virus was harvested. The lysate was transferred to a 50 mL conical tube (SPL Life Science, Pocheon, Republic of Korea) and centrifuged at 500× *g* for 10 min at 4 °C. The supernatant containing the viral particles was collected, filtered through a 0.45 μm filter, and stored at −80 °C until further use. The viral copies were determined using real-time PCR as described in our previous study [11]. For vaccine preparation, a cultured virus containing more than 1 × 10^10^ RSIV genome copies/mL was used. Vaccines were prepared as described in a previous study, with some modifications [12]. Briefly, after thawing the cultured virus at 25 °C, formalin was added to achieve a final concentration of 0.1% (*v*/*v*), mixed thoroughly, and kept at 37 °C for 4 h. Complete inactivation was confirmed by injecting the formalin-inactivated virus into naïve rock bream (n = 50). No mortality was observed in the fish for 30 days post-injection, confirming successful inactivation. The pH of the prepared vaccines was measured and found to be in the range of 7.32–7.41. After confirming the complete inactivation, the vaccines were stored at 2–8 °C until further use.

### 2.3. Fish and Immunization

Healthy rock bream acquired from a local hatchery were stocked at different commercial fish farms in southern offshore Korea. At fish farm A, a land-based tank aquaculture facility, 41,500 individuals (average body weight: 18.4 ± 4.24 g) were counted. Meanwhile, at fish farm B, located in offshore marine cages, the total count was 61,800 individuals (average body weight: 21.1 ± 3.92 g). Each group was acclimatized for over 2 weeks and fasted for one day prior to the vaccination procedure. Additionally, before vaccination, five rock breams were randomly selected and examined for RSIV using nested polymerase chain reaction [12]. Vaccination was performed by intraperitoneal (i.p.) injection of 0.1 mL of the vaccine on 31,100 fish from farm A and 30,700 fish from farm B. The remaining fish in each cage were used as controls. At fish farm A, a land-based tank aquaculture facility, the immunized fish were cultured at 23–25 °C for 7 days post-vaccination and then transferred to offshore marine cages. At fish farm B, owing to on-site conditions, vaccination was performed 5 weeks later than at fish farm A.

### 2.4. Efficacy Test in Field Trial

Field-scale experiments were conducted at commercial fish farms A and B in offshore marine cages. To evaluate the status of viral infection in the field and compare the efficacy of the vaccine over different vaccination periods, fish in the field were sampled and transported to the laboratory every three weeks. Subsequently, in cases where RSIV infection was detected in fish farms, sampling was performed to confirm ongoing infection; however, artificial inoculation was not conducted. Conversely, on farms where no RSIV infection was detected, fish were sampled and transported to the laboratory to evaluate vaccine efficacy through artificial inoculation (Figure 1). Throughout the field test, daily mortality rates and clinical symptoms were recorded until November 2020, during which field infections were not persistent. The final cumulative mortality rate was calculated using the following formula:
$$\mathrm{Cumulative}\;\mathrm{mortality}\left(\%\right)=\frac{\left(\mathrm{Initial}.\;\mathrm{of}\;\mathrm{stocked}\;\mathrm{fish}-\mathrm{no}.\;\mathrm{of}\;\mathrm{fish}\;\mathrm{surviving}\;\mathrm{at}\;\mathrm{the}\;\mathrm{end}\right)}{\mathrm{Initial}.\;\mathrm{of}\;\mathrm{stocked}\;\mathrm{fish}}\times 100$$

### 2.5. Efficacy under Laboratory Conditions

The transported fish were acclimatized to 25 °C by increasing or decreasing the temperature (1–2 °C/day) before challenge. Additionally, 3–5 of these fish were subjected to blood sampling, weight measurements, and organ extraction. After reports of RSIV infections along the southern coast, each group was observed for over 7 days. During this period, extracted organs were subjected to PCR testing to confirm non-infection. Simultaneously, individuals showing no clinical symptoms were selected for the challenge test to evaluate the efficacy of the vaccine, where they were inoculated with IVS-1 by i.p. injection (1.0 × 10^4^ RSIV genome copies/0.1 mL/fish). Post-challenge, each group of fish was maintained at 25 °C in the 100 L aquarium tank with 80% water exchanged seawater and aeration (Appendix A), and fed commercial feed at 1–3% of body weight daily until the experimental period. The tanks were monitored daily for clinical signs of disease and mortality.

### 2.6. Growth Performance and Spleen Index

Fish from each transported group were randomly selected, weighed (in grams), and then dissected to extract the spleen. After measuring the spleen weight (in milligrams), the spleen index was calculated according to the following formula.
Spleen index=[Spleen weight/Fish weight (g)]×1000

### 2.7. Neutralizing Antibodies in Serum

Neutralizing antibodies in the serum were detected as described previously, with some modifications [11]. Briefly, blood samples for neutralization testing were collected from the caudal vein of each rock bream every three weeks. Serum (3 individuals of each group) was then separated by centrifuging at 4 °C at 5000× *g* for 15 min. Following complement inactivation, the serum was diluted ten times with L-15 medium and subsequently subjected to a 2-fold serial dilution. Each of dilutions (100 μL) were combined with an equal volume of IVS-1 (1 × 10^6^ RSIV genome copies/100 μL) and incubated at 25 °C for one hour within a 96-well plate. The mixture containing neutralized IVS-1 was then inoculated into dwarf gurami fin (DGF) cells [13], which were cultured in 24-well tissue culture plates for five days. These infected cells were further incubated for seven days at 28 °C to monitor the cytopathic effects (CPE).

### 2.8. Statistical Analysis

All graphs were generated using the GraphPad Prism software (version 10.0; GraphPad Software, Inc., Boston, MA, USA). The mortality of the vaccinated and control groups under laboratory trials was calculated to RPS and analyzed using a log-rank (Mantel–Cox) test. Specific antibody levels were compared using two-way analysis of variance with Tukey’s multiple comparison test. All data were considered statistically significant at *p* < 0.05. RPS was calculated using the cumulative mortality (RPS = (1 − mortality of vaccinated group/mortality of control group) × 100).

## 3. Results

### 3.1. Growth Performance and Spleen Index

Two separate field tests were conducted to evaluate the efficacy of formalin-killed vaccines in marine cage-cultured rock bream in July 2020 (farm A) and August 2020 (farm B). After vaccination, fish from both farms showed continuous growth throughout the field trial period, and there were no significant differences in body weight between the vaccinated and control groups (Figure 2A,B).

Additionally, to clinically assess RSIV infection based on relative weight changes in the spleen, which is a major clinical symptom of RSIV, we measured the spleen index to confirm spleen enlargement. At farm A, over the 18-week period, the average spleen index was 0.88 ± 0.21 in the vaccinated group and 0.83 ± 0.13 in the control group, indicating that the spleen index remained around 1.0 (Figure 2C). Conversely, at farm B, the spleen index was 0.74 (vaccinated) and 0.69 (control) at 3 weeks post-vaccination (wpv). However, at 9 wpv, the spleen index increased to 1.91 in the vaccinated group and 2.11 in the control group, indicating significant spleen enlargement. By 12 wpv, when mortality ceased and the spleen index of the vaccinated group was 0.73, while that of the control group was 1.30 (Figure 2D).

### 3.2. Efficacy of the Vaccine in Laboratory Conditions

Although RSIV infections were reported in surrounding marine cage farms at 8 wpv, no RSIV infection was identified at farm A during the vaccine efficacy analysis (Table 1). Consequently, samples collected from week 9 onwards underwent acclimatization at 25 °C for over one week to confirm their non-infection status before artificial infection. In the subsequent artificial injection experiment, the cumulative mortality rate of all the control groups from fish farm A reached 100%. In contrast, the vaccinated group showed cumulative mortality rates of 40.0%, 36.4%, 33.3%, 21.1%, 20.0%, and 25.0% at 3, 6, 9, 12, 15, and 18 wpv, respectively, resulting in RPS ranging from 60% to 80% (Figure 3, Appendix A). These findings demonstrate the protective efficacy of the vaccine up to 18 wpv.

For fish farm B, at 3 wpv, the fish were acclimatized at 25 °C for one week before viral infection analysis. Artificial infection was induced after confirmation of non-infection. The vaccinated group exhibited a cumulative mortality rate of 33.3%, whereas that of the control group reached 100.0%, resulting in an RPS of 66.7%. This suggests that the fish at farm B exhibited protective efficacy against the vaccine (Figure 4, Appendix A).

### 3.3. Antibody Titers in Serum from Vaccinated Rock Bream

Specific antibodies against IVS-1 in the serum of vaccinated rock bream were determined using the DGF cell line in 96-well plates. For the serum samples isolated from the fish at farm A, the geometric mean titer (GMT) showed 120.0 ± 56.7 in the vaccinated group at 3 wpv. The control fish had a GMT of 10. Between 6 and 18 wpv, the GMT ranged from 133.3 to 240, indicating that the vaccine played a role in inducing antibody production and persisted up to 18 weeks (Figure 5A). In farm B, the GMT at 3 wpv was 133.3 ± 46.2. However, at 6 and 9 weeks, neutralizing antibody analysis was not feasible because of RSIV infection. Subsequently, in fish at 12 wpv where the virus was not detected, the GMT showed results of 320 ± 277.1 and 266.7 ± 92.4 for the vaccinated and control groups, respectively (Figure 5B).

### 3.4. Vaccine Efficacy in a Field Outbreak

In contrast, at farm B, from 5 wpv, when viral infections were confirmed, further transportation to the laboratory and artificial infections were not conducted. Instead, an on-site efficacy analysis was conducted based on the number of dead fish. The water temperature ranged from 17 °C to 25 °C during field tests, with mortality beginning at 6 wpv in the control group and at 7 weeks in the vaccinated group, when the water temperature was in the range of 21 to 23 ± 1 °C (Figure 6). This was recorded until 12 wpv, with cumulative mortality in the control and vaccinated groups of 80.3% and 36.4%, respectively, showing an RPS of 54.67%. The dead fish exhibited severe lethargy, anemia, ocular hemorrhage, and splenic enlargement, which are typical signs of this disease. Additionally, the surviving fish from farm B at 12 weeks post-vaccination did not experience mortality despite being exposed to viral concentrations ten times higher than those used in the laboratory challenge experiments. On farm A, where no RSIV infections were reported in the field, there was no significant mortality due to the disease during the field trial period. The cumulative mortality rates in the vaccinated and control groups were 2.09% and 3.08%, respectively.

## 4. Discussion

RSIV, which causes RSIVD in marine fishes, poses a serious danger to aquaculture. Instead of drugs, vaccination has been demonstrated to be a safe method for controlling infectious diseases [14]. A formalin-killed vaccine against RSIVD is effective and commercially available in Japan [15,16]. However, rock bream, a fish species highly susceptible to RSIV-subtype II *megalocytivirus*, has been extensively affected by RSIV in South Korea, showing 60–100% cumulative mortality under artificial infection [3,6,17]. According to the MCP analysis study by [1], the Ehime-1 strain used in commercial vaccine is RSIV-subtype I, which is different from the RSIV-subtype II. Additionally, this commercial vaccine has shown limited effectiveness against the genus *Oplegnathus* [1,18]. Although trials are currently being conducted to determine the efficacy of DNA and rock bream vaccines [19,20], in field trials characterized by various factors such as co-infections with various pathogens and fluctuations in water temperature and the environment, vaccine efficacy has been shown to be less effective than in the laboratory. As a result, there is a highlighted need for an effective vaccine targeting RSIV subtype-II that ensures high efficacy under field conditions.

Our previous study evaluated the efficacy of the vaccine by considering various factors such as virus concentration, inactivation methods, adjuvants, and vaccination temperature [11]. Based on the promising results observed at the laboratory level, the current study aimed to assess the applicability of the vaccine under field conditions. Consequently, we conducted this study on 103,200 rock bream individuals at two farms located along the southern coast of South Korea.

Before evaluating the efficacy of vaccines using the mortality rate of experimental fish, it is essential to confirm the safety of the vaccine, any side effects caused by vaccination, and the health and infection status of the fish. These factors are crucial for understanding the characteristics of field infections. In both farms A and B, consistent growth was observed in vaccinated and control groups, indicating that the vaccine did not cause growth retardation. Notably, PCR and spleen index analyses provided clear evidence of the effect of the vaccine on disease prevention. At farm A, the fish sampled over the 18 weeks tested negative for the virus by PCR (Table 1), and the spleen index values remained around 1.0 (Figure 2C), indicating a healthy status without infection. These results are similar to the spleen index values for non-infected fish in a previous study [21]. In contrast, at farm B, RSIV infection was identified at 5 wpv (Table 1), and the spleen index values increased to around 2.0 at 9 wpv, indicating spleen enlargement. Of note, a critical spleen index threshold of 2 due to RSIV infection is closely related to mortality in rock bream [21]. Therefore, these results suggested that vaccinated fish possess an enhanced ability to recover from infections, which is a critical factor in maintaining fish health and reducing mortality rates in aquaculture.

Additionally, the RPS, the primary parameter for evaluating aquatic vaccine efficacy, measures the level of protection provided by the vaccine. According to our previous study, the efficacy of an RSIV vaccine is proportional to its concentration [12]. As expected, a notable increase in RPS was observed in groups that were vaccinated in this study. For farm A, where RSIV infection did not occur on-site, the RPS ranged from 60% to 80% over a period of 3 to 18 weeks (Appendix A). This indicated that the protective efficacy of the vaccine, observed from 3 wpv onwards, persisted for at least 18 weeks (Figure 3). The results from farm B also demonstrated a similar efficacy, with approximately 19,520 surviving fish in the vaccinated group and 6100 fish in the control group during the observation period of approximately 12 weeks. The final cumulative mortality rates were 36.43% and 80.32% with an RPS of 54.67%, respectively. Although this RPS of 54.67% appeared relatively lower than the RPS observed at farm A, it still indicated substantial protection offered by the vaccine under field conditions. Although sampling was only performed up to 18 wpv at farm A and 12 wpv at farm B, considering the virus infection conditions in the field, it is likely that the vaccine efficacy could persist beyond these periods. In particular, although squalene and aluminum hydroxide adjuvants showed great effectiveness in a previous study [11], they were not used in large-scale field vaccinations due to economic constraints. Nevertheless, the vaccine demonstrated sufficient efficacy in the field conditions. Although the field vaccination was conducted at two farms, natural infections were only observed at farm B. However, during the field trial, there were continuous reports of RSIV infections in nearby farms at farm B. Furthermore, the vaccinated and control groups were placed close to each other in marine cages, meaning that virus release from the control group could lead to additional pathogen exposure in the vaccinated group. Therefore, despite field vaccine efficacy being confirmed at only one farm, considering these factors, it can be concluded that the vaccine’s effectiveness was adequately demonstrated. Nevertheless, further research in various environments is necessary to comprehensively evaluate vaccine efficacy.

According to this result, a likely explanation for the potentially reduced performance in the field compared to what is seen in laboratory trials is the constant presence of various heterologous pathogens in field populations. Sea lice are present in the fish population of Korea and can affect the growth, reproduction, and survival of the fish they infest [22]. In addition, the indirect effects of immunosuppression allow other pathogens to infect fish, leading to associations or co-infections with other pathogens [23]. Farm B, which suffered annual losses due to coinfection with RSIV and sea lice, was no exception in this study.

Another factor to consider is the difference in water temperature between farm B, which is a marine cage farm, and farm A, which has land-based tanks. Following vaccination, farm B maintained water temperatures ranging from 20 to 24 °C, while farm A allowed for temperatures between 23–25 °C. Water temperature is one of the most important factors that determine vaccine efficacy in fish [24,25]. In previous studies, the effect of temperature on vaccine efficacy was tested by shifting the water temperature up (from 21 °C to 25 °C) and down (from 25 °C to 21 °C) following vaccination [11]. The results indicated a significant difference in vaccine efficacy in the shifted-down group. Additionally, in the same study, the group maintained at 25 °C showed similar or higher vaccine efficacy compared to the shifted-up group. Thus, although the RPS at farm B was expected to be lower due to temperature variations, the RPS at farm B (66.7%) was higher than that at farm A (40.0%) at 3 wpv, suggesting that several factors may influence vaccine efficacy in the field.

According to the “Manual for Quality Control of Fish Vaccines” of the NIFS, which is the quality management guideline for fish vaccines in Korea, the criterion for determining vaccine efficacy states that the survival rate in the vaccinated group should be more than 40% higher than that in the control group, where at least 60% mortality was observed in the control group due to the same concentration of virus. The RPS observed in this study greatly exceeded this criterion, indicating the remarkable effectiveness of the vaccine in the field and providing significant insights into the characteristics of infection under field conditions. However, given the significant annual fluctuations in the water temperature in the field, further evidence of vaccine efficacy under diverse environmental conditions is required to ensure reproducibility in the field. Although this study did not include a comparison with commercially available vaccines, further study is necessary to conduct cross-protection studies among different subtypes of RSIV.

By associating it with protection, the antibody response is a frequently used metric to assess the effectiveness of fish immunization [26]. In the field, vaccinated fish showed the formation of neutralizing antibodies at 3 weeks post-vaccination, with GMTs of approximately 120.0 ± 56.6 and 133.3 ± 46.2 at farm A and farm B, respectively. At farm A, GMTs remained high until 18 wpv in vaccinated fish, whereas the control groups at the same time point exhibited GMTs of less than 10, indicating the absence of antibody formation. In farm B, where viral infections occurred in the field, surviving fish exhibited neutralizing antibody titers post-infection, with GMTs of approximately 320 ± 277.1 in the vaccinated group and 266.7 ± 92.4 in the control group. These findings indicated the presence of neutralizing antibodies in all surviving fish following viral infection in the field.

## 5. Conclusions

Although no RSIV outbreaks occurred during the field trial at farm A, the challenge test under laboratory conditions indicated that the vaccinated fish were effectively protected against RSIV for at least 18 weeks. Furthermore, the results of the field trial at farm B demonstrated the efficacy of the vaccination in providing protection. These results are supported by a similar trend in neutralizing antibody levels detected in serum samples collected from experimental fish.

## Figures and Tables

**Figure 1 vaccines-12-00680-f001:**
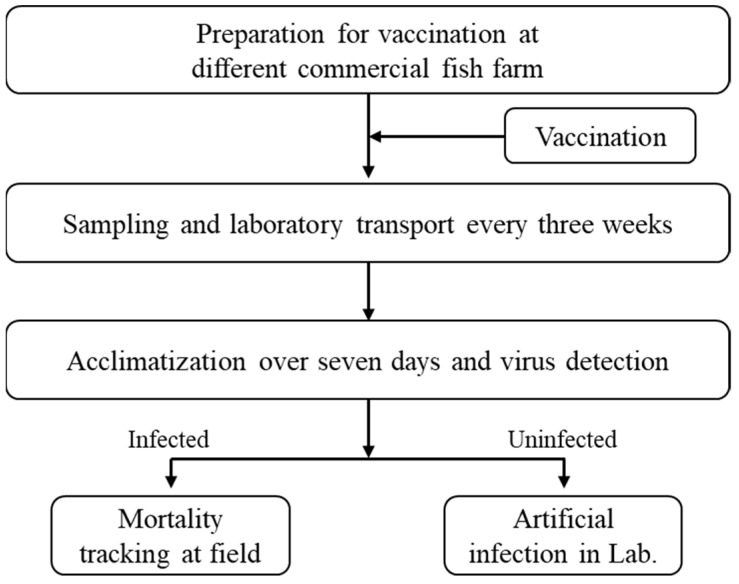
Schematic illustration for analysis of vaccine efficacy in laboratory and field conditions.

**Figure 2 vaccines-12-00680-f002:**
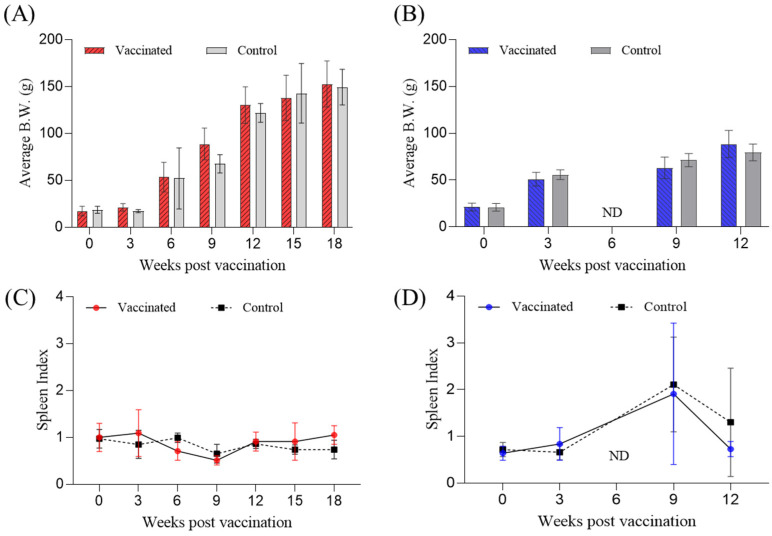
Comparative analysis of average body weight (**A**,**B**) and spleen index (**C**,**D**) of rock bream in vaccinated and control groups throughout the 18-week (farm A) and 12-week (farm B) trials. (**A**,**C**) data from farm A, representing changes in body weight and spleen index, respectively. (**B**,**D**) Display similar data for farm B. ND: not done.

**Figure 3 vaccines-12-00680-f003:**
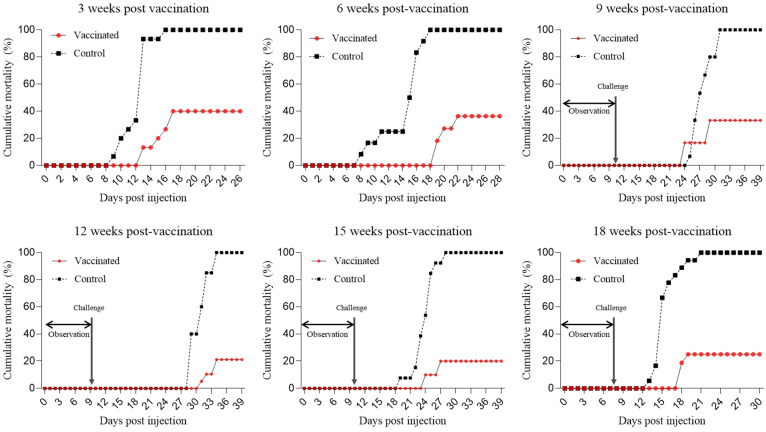
Overall cumulative mortality of rock bream in the vaccinated (●) and control (■) groups following an intraperitoneal challenge at 25 °C with IVS-1 (10^4^ copies/mL) at 3, 6, 9, 12, 15, and 18 weeks post-vaccination from farm A. Following reports of RSIVD outbreaks in the surrounding marine area after 8, fish were kept at 25 °C for over 7 days to confirm non-infection before challenge injection. After reports of RSIVD outbreaks in the surrounding marine area (after 8 wpv), fish were observed for over 7 days at 25 °C to confirm the absence of infection before the challenge test. A significant difference was noted among the vaccinated and control groups with *p*-values of <0.0001, <0.0001, 0.0002, 0.0006, <0.0001, and <0.0001 at 3, 6, 9, 12, 15, and 18 weeks post-vaccination, respectively.

**Figure 4 vaccines-12-00680-f004:**
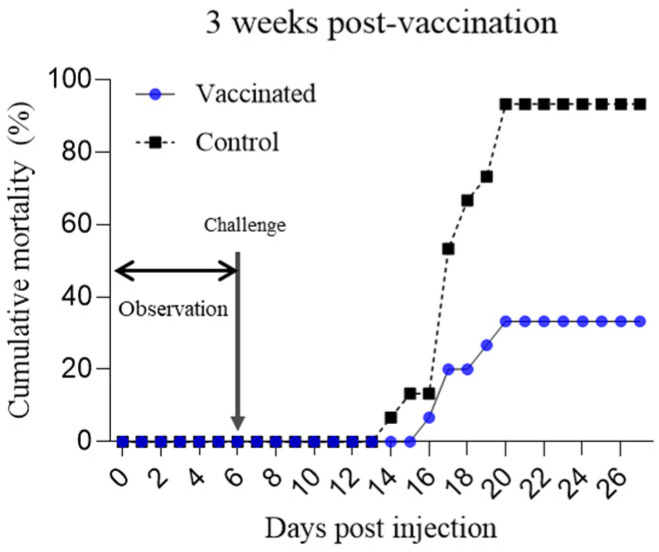
Cumulative mortality of rock bream in the vaccinated (●) and control (■) groups following an intraperitoneal challenge at 25 °C with IVS-1 (10^4^ copies/mL) at 3 weeks post-vaccination from farm B. Fish were observed for 6 days to confirm non-infection before challenge injection. A significant difference was noted among the vaccinated and control groups with *p*-values of 0.0002.

**Figure 5 vaccines-12-00680-f005:**
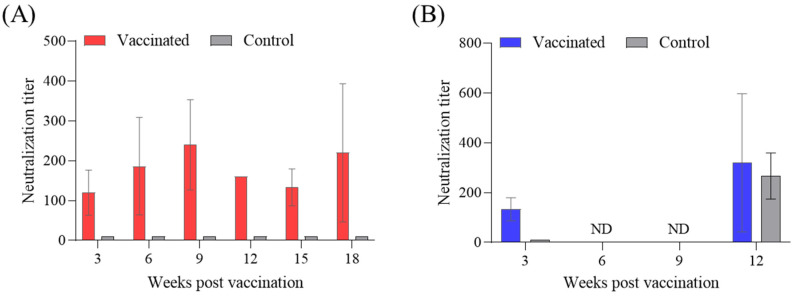
Levels of neutralizing antibody titer in the serum of rock bream. The levels of antibodies were examined from the serum of vaccinated and control, with three fish per group throughout the 18-week (farm A) and 12-week (farm B) trials. ND: not done.

**Figure 6 vaccines-12-00680-f006:**
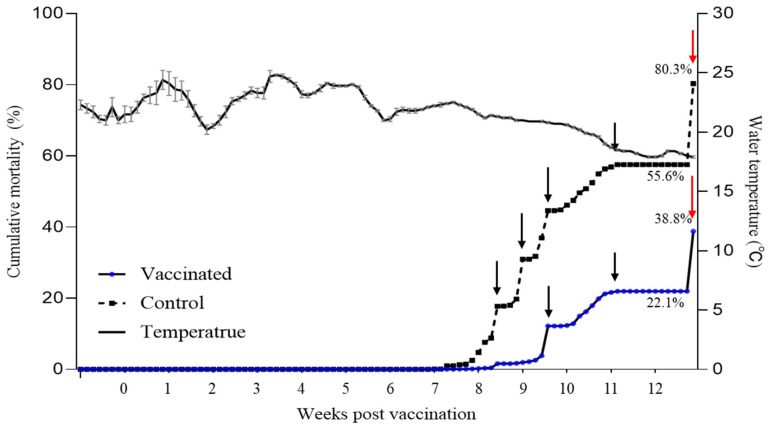
Cumulative mortality and seawater temperature data for vaccine efficacy using rock bream in the field trial. Mortality was assessed by counting the number of dead fish floating on the surface from farm B. Black arrows indicate the total number of dead fish found at the bottom of the net by lifting the cage net. Red arrows show the final cumulative mortality. The final cumulative mortality rate was calculated by subtracting the count of surviving individuals in the net-pen from the original stock.

**Table 1 vaccines-12-00680-t001:** Analysis of RSIV infection status in field fish by PCR.

Location	Group	Weeks Post-Vaccination						
		0	3	6	9	12	15	18
Farm A	Vaccinated	0/3 ^a^	0/5	0/3	0/5	0/3	0/3	0/3
	Control		0/5	0/3	0/3	0/3	0/3	0/3
Farm B	Vaccinated	0/3	0/3	3/3 ^b^	2/8	0/3	NT ^c^	NT
	Control		0/3	3/3 ^b^	4/6	0/3	NT	NT

^a^: no. of positive samples/analyzed samples by PCR; ^b^: In farm B, dead fish were used at 5 wpv rather than 6 wpv for analyzation; ^c^: not tested.

## Data Availability

The datasets used in this study are available from the corresponding authors upon reasonable request.

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
