# Peer review of "Evaluation of Formalin-Inactivated Vaccine Efficacy against Red Seabream Iridovirus (RSIV) in Laboratory and Field Conditions"

_vaccines, 2024, doi:10.3390/vaccines12060680_

Round 1
Reviewer 1 Report
Comments and Suggestions for Authors
In this study, the authors evaluated the efficacy of a formalin-killed vaccine against RSIV in rock bream under laboratory and field conditions. This study demonstrates that the formalin-killed vaccine effectively prevents RSIV in cage-cultured rock bream under both laboratory and field conditions. However, some issues should be addressed before publication.
1. Line 32, 44, 287: (Kurita and Nakajima, 2012), (Jung et al., 2018), Shin et al. (2024) reported, the authors should the reference format.
2. line 83-87: “Vaccine preparation”, the formalin should be removed after inactivation.
3. line 122-131: “Efficacy under laboratory conditions”, how many fish for challenge test? Did the water temperature maintain at 25°C during the challenge test? The authors should provide more details?
4. line 180-182, “RSIV infections were reported in surrounding marine cage farms”, the fish were not infected by RSIV at farm A?
5. In Fig. 3 and Fig. 4, Did the water temperature maintain at 25°C during the challenge test? Fig. 3 means the water temperature maintain at 25°C at the first 10 days? It's easy to misunderstand of Fig. 3 and Fig. 4.
6. line 231-234: Asterisks (**) indicate significant differences at p < 0.01. The authors should add in the manuscript.
7. line274-299: repeat of section “Result”, not suitable for Discussion.
Author Response
# Reviewer 1
Q1. Line 32, 44, 287: (Kurita and Nakajima, 2012), (Jung et al., 2018), Shin et al. (2024) reported, the authors should the reference format.
Response: We agree with your comment. As per your comment, we revised the reference format throughout the MS.
Q2. line 83-87: “Vaccine preparation”, the formalin should be removed after inactivation.
Response: We are grateful for your thorough review and valuable feedback. We conducted safety tests on the concentration of formalin used for inactivation, and no mortality was observed in any of the vaccinated fish with 0.1% (v/v) formalin-inactivated virus. We have added this information to the Materials and Methods section to clarify the safety and efficacy of the formalin-inactivated vaccine (L96-100).
Q3. line 122-131: “Efficacy under laboratory conditions”, how many fish for challenge test? Did the water temperature maintain at 25°C during the challenge test? The authors should provide more details?
Response: We sincerely appreciate your insightful comments and the effort you put into reviewing our manuscript. Considering your comments, we have revised the Materials and Methods section to include detailed information about the number of fish used for the challenge test and the maintenance of water temperature at 25°C during the challenge test (L137-145). Additionally, we have provided a supplemental table to present this information more comprehensively (Table S1). Thank you for your suggestion, which has helped improve the completeness of our manuscript.
Q4. line 180-182, “RSIV infections were reported in surrounding marine cage farms”, the fish were not infected by RSIV at farm A?
Response: Thank you for your constructive comments and suggestions on our manuscript. Fish at Farm A were confirmed to be uninfected by RSIV through PCR testing (Table 1). Considering your comments, we have revised the MS to clarify this point and ensure greater clarity (L193-194). Thank you for bringing this to our attention.
Q5. In Fig. 3 and Fig. 4, Did the water temperature maintain at 25°C during the challenge test? Fig. 3 means the water temperature maintain at 25°C at the first 10 days? It's easy to misunderstand of Fig. 3 and Fig. 4.
Response: We appreciate your detailed review and valuable insights. As per your comments, there was potential for misunderstanding regarding the maintenance of water temperature at 25°C in Fig. 3 and Fig. 4. After transportation in the laboratory, rearing-water temperatures were maintained at 25°C during the challenge test. Considering your comments, to address this issue, we have revised both figures to clearly indicate the temperature conditions throughout the challenge test. Thank you for highlighting this important aspect, and we have made the necessary amendments to our manuscript to improve clarity.
Q6. line 231-234: Asterisks (**) indicate significant differences at p < 0.01. The authors should add in the manuscript.
Response: Thank you for your constructive comments. Upon statistical analysis, we found that the vaccinated group showed a significant difference compared to the control group during the experiment periods. However, we believe it is more appropriate to analyze the statistical significance within the vaccinated group across different weeks post-vaccination (wpv). In this analysis, no statistical difference was observed. Considering you and other reviewers’ comments, we have removed the asterisk (*) indicating significance. We appreciate your attention to this detail and have revised Figure 5.A accordingly.
Q7. line274-299: repeat of section “Result”, not suitable for Discussion.
Response: We agree with your observation that this section contained repeated information from the "Results" section and was not suitable for the "Discussion" section. We have thoroughly revised the "Discussion" section to eliminate this repetition and to ensure that it appropriately interprets and contextualizes the findings of our study (L300-312). Thank you for your constructive feedback.

Reviewer 2 Report
Comments and Suggestions for Authors
This work mainly evaluated the efficacy of the formalin-inactivated red seabream iridovirus vaccine, including in laboratory and field experimental conditions. The results of the study provided basic data for the further application of the vaccine. However, there are some issues that the authors need to consider:
1. The main purpose of this study is to evaluate the efficacy of the vaccine, so the preparation of the vaccine should be clear and fixed in the early stage. In the methods section, the virus propagation and the vaccine preparation were not shown to this extent.
2. The field experiment was carried out at only two farms, and the conditions of the two farms were different. Especially for farm B with natural RSIV infection, the RPS value obtained from a single infection event is only for reference.
3. Line 225: The results showed that protection with this vaccine lasted for at least 18 weeks. This may be because the monitoring only lasted for 18 weeks.
4. Figure 5.A: Why there was a significant increase in geometric mean titer at 18 wpv.
5. Lines 320-326: The effect of temperature can be tested in the laboratory. Is there any relevant previous data?
Author Response
Q1. The main purpose of this study is to evaluate the efficacy of the vaccine, so the preparation of the vaccine should be clear and fixed in the early stage. In the methods section, the virus propagation and the vaccine preparation were not shown to this extent.
Response: We agree with your insightful comments. As per your comments, to ensure clarity in the vaccine manufacturing process, we have revised the Materials and Methods section. Specifically, the sections previously titled “Cell culture and Virus propagation” and “Vaccine preparation” have been consolidated and revised to “Cell culture” and “Virus propagation and vaccine preparation” (L83-101). This reorganization provides a clearer and more detailed description of the entire process.
Q2. The field experiment was carried out at only two farms, and the conditions of the two farms were different. Especially for farm B with natural RSIV infection, the RPS value obtained from a single infection event is only for reference.
Response: Thank you for your valuable comments and suggestions. As you pointed out, the field data is primarily based on the results from Farm B, where natural RSIV infection occurred. We recognize that this is a limitation of our study due to the inherent constraints of field experiments. To address this, we conducted sampling every three weeks to evaluate vaccine efficacy until natural infections were reported on-site. Specifically, at farm B, virus release from mortality in the control group could have served as an additional infection source for the vaccinated group. This allowed us to assess the vaccine's effectiveness under these challenging conditions. However, we acknowledge the limitation you highlighted and have included this discussion in the revised Discussion section (L331-339). We appreciate your insightful comment and have made the necessary amendments to our manuscript.
Q3. Line 225: The results showed that protection with this vaccine lasted for at least 18 weeks. This may be because the monitoring only lasted for 18 weeks.
Response: We sincerely appreciate your insightful comments and the effort you put into reviewing our manuscript. Regarding this point, we monitored the fish for up to 30 weeks post-vaccination at Farm A, and our findings confirmed that the vaccine's efficacy persisted throughout this period. However, to maintain a conservative interpretation of the results, we initially described the efficacy duration as 18 weeks in the original MS, which was the minimum observed period for which we had robust data. Considering your comments, we have included additional information in the Discussion section to reflect this extended monitoring period and to clarify the sustained efficacy of the vaccine (L325-327)."
Q4. Why there was a significant increase in geometric mean titer at 18 wpv
Response: We sincerely appreciate your insightful comments and the effort you put into reviewing our manuscript. Upon statistical analysis, we found that the vaccinated group showed a significant difference compared to the control group during the experiment periods. However, we believe it is more appropriate to analyze the statistical significance within the vaccinated group across different weeks post-vaccination (wpv). In this analysis, no statistical difference was observed, so we have removed the asterisk (*) indicating significance. We appreciate your attention to this detail and have revised Figure 5.A accordingly.
Q5. Lines 320-326: The effect of temperature can be tested in the laboratory. Is there any relevant previous data?
Response: We are grateful for your thorough review and valuable feedback. Considering your comments, we have added relevant information regarding previous studies on the effect of temperature to the Discussion section (L348-359). This addition provides context and supports our findings by referencing prior research on temperature's impact on vaccine efficacy.

Reviewer 3 Report
Comments and Suggestions for Authors
This report presents the results of the evaluation of the protectivity of the inactivated vaccine against Red seabream iridovirus. This vaccine was developed and tested on a small group of fish in a previous study (Kwon et al., 2020). Here, the authors used a larger number of fish from two different farms to confirm their conclusions about the prospects of using the vaccine candidate.
As for the methods and approaches of the study, they repeat the previous research (Kwon et al., 2020). However, the number of experimental fish in this research is amazing, which allowed the authors to get statistically significant results.
However, on the other hand, it seems to me that the work lacks fundamentally new data. The conclusions of the article repeat the conclusions of the previous study. Nevertheless, I believe that the work can be published in Vaccines.
After first reading this report, I had a lot of questions about the design of the experiments. However, I found answers to many questions in the previous article (Kwon et al., 2020). In this regard, I would like to suggest that the authors make a number of additions to the text to improve it.
1. This study is based on and partially repeats the previous study of the authors (Kwon et al., 2020). Information about the results should be added to the Introduction section in order to understand the logic of the new experiment.
2. The authors indicate in the text that licensed inactivated vaccines exist in Japan, including the Biken vaccine, and note their low effectiveness. At the same time, it is not clear why the authors propose the same technology for creating a vaccine and why they expect a higher efficiency result. This information should be added to the Introduction and possibly Discussion section.
3. In their previous work, the authors evaluated the effectiveness of vaccine candidates when adding various adjuvants. One of the conclusions of the study was that the addition of an adjuvant based on squalene and aluminum hydroxide showed the greatest effectiveness. Why didn't the authors use adjuvants in this work? This should be discussed in detail in the Discussion section.
4. The most significant comment on the design of the study. Why did the authors not include another control group where fish were immunized with a licensed formalin inactivated vaccine/vaccines. Within a single experiment, it would be possible to demonstrate the difference in the protective properties of the vaccine candidate and existing vaccines.
5. It is necessary to add several keywords.
6. Line 39-40. The association between antibiotics and viral infections is not clear.
7. The format of references 7, 19 and 20 is probably different from the general one.
Author Response
Q1. This study is based on and partially repeats the previous study of the authors (Kwon et al., 2020). Information about the results should be added to the Introduction section in order to understand the logic of the new experiment.
Response: Thank you for your insightful comments and suggestions on our manuscript. We appreciate your valuable comment. In our previous study (Kwon et al., 2020), we evaluated the RSIV vaccine efficacy based on several factors including antigen-dose, inactivation methods, advent types and water temperature. Among the factors tested, rock bream vaccinated with a high-dose, ultracentrifuged megalocytivirus vaccine (Ultra HSCMV, 7.0 × 1010 copies/mL) showed significantly higher survival, with only 10% cumulative mortality, compared to those given relatively low-dose vaccines (1.0 × 1010 copies/mL, 1.0 × 109 copies/mL and 1.0 × 108 copies/mL), which had cumulative mortality rates of 48.3 ± 7.6%, 75.0 ± 5.0%, and 100.0 ± 0.0%, indicating an antigen dose-dependent vaccine efficacy. Considering your comments, we have revised the Introduction section to include information about the results and findings of our previous study (Kwon et al., 2020). This addition helps to clarify the rationale and context for the new experiment presented in this study (L55-67).
Q2. The authors indicate in the text that licensed inactivated vaccines exist in Japan, including the Biken vaccine, and note their low effectiveness. At the same time, it is not clear why the authors propose the same technology for creating a vaccine and why they expect a higher efficiency result. This information should be added to the Introduction and possibly Discussion section.
Response: We value your feedback and the effort you put into reviewing our manuscript. Thank you for your insightful comment. The Biken vaccine available in Japan uses the RSIV subtype-1 strain, which, according to phylogenetic analysis, differs from the RSIV subtype-2 strain that causes RSIV infections in Korea. This difference likely accounts for the lower effectiveness of the Biken vaccine in Korea. We have revised the Introduction and Discussion sections of the manuscript to include this important distinction and to explain why our vaccine, targeting RSIV subtype-2, is expected to demonstrate higher efficacy (L46-54, 279-290).
Q3. In their previous work, the authors evaluated the effectiveness of vaccine candidates when adding various adjuvants. One of the conclusions of the study was that the addition of an adjuvant based on squalene and aluminum hydroxide showed the greatest effectiveness. Why didn't the authors use adjuvants in this work? This should be discussed in detail in the Discussion section.
Response: Thank you for your constructive comments and suggestions on our manuscript. Due to economic constraints associated with vaccine production, we did not use adjuvants such as squalene and aluminum hydroxide in this study. However, we observed that the vaccine was still sufficiently effective without the addition of these adjuvants. We acknowledge that incorporating adjuvants could enhance efficacy further and will consider this in future research. Considering your comments, we have added this explanation to the Discussion section to address your comment in detail (L 327-331).
Q4. The most significant comment on the design of the study. Why did the authors not include another control group where fish were immunized with a licensed formalin inactivated vaccine/vaccines. Within a single experiment, it would be possible to demonstrate the difference in the protective properties of the vaccine candidate and existing vaccines.
Response: We appreciate your detailed review and valuable insights. We acknowledge the importance of including a control group immunized with a licensed formalin-inactivated vaccine to compare the protective properties directly. However, the distribution of currently available vaccines is challenging, and the procedures required for field vaccination are extensive, which prevented us from including this control group in the present study. We recognize the value of your suggestion and will consider incorporating licensed vaccines in further study to provide a more comprehensive comparison (L369-371). Thank you for your insightful comment.
Q5. It is necessary to add several keywords.
Response: Thank you for your comments. Based on your comments, we have added keywords (L24)
Q6. Line 39-40. The association between antibiotics and viral infections is not clear.
Response: Thank you for your comments. Considering your comments, to avoid confusion, we have focused on the vaccination except for antibiotics. We have revised the manuscript to clarify the association between antibiotics and viral infections to ensure greater clarity (L39-45). Thank you for highlighting this point.
Q7. The format of references 7, 19, and 20 is probably different from the general one.
Response: We agree with your comment. As per your comment, we revised the reference format throughout the MS.

Round 2
Reviewer 2 Report
Comments and Suggestions for Authors
The authors gave detailed responses to all the comments. Although there are some deficiencies, it does not affect the integrity and scientific value of this manuscript.
I have no more comments.
Reviewer 3 Report
Comments and Suggestions for Authors
The authors made all the necessary changes to the text of the MS and answered the questions. A revised version of the MS may be published in the Vaccine.